# Recent Advances and Future Directions in Clinical Management of Head and Neck Squamous Cell Carcinoma

**DOI:** 10.3390/cancers13020338

**Published:** 2021-01-18

**Authors:** Jameel Muzaffar, Shahla Bari, Kedar Kirtane, Christine H. Chung

**Affiliations:** 1Moffitt Cancer Center, Department of Head and Neck-Endocrine Oncology, Tampa, FL 33612, USA; Jameel.muzaffar@moffitt.org (J.M.); Kedar.Kirtane@moffitt.org (K.K.); 2Hematology Oncology Fellow, Moffitt Cancer Center, Tampa, FL 33612, USA; Shahla.Bari@moffitt.org

**Keywords:** head and neck squamous cell carcinoma, human papillomavirus, smoking, cisplatin, radiation therapy, induction chemotherapy, immunotherapy, programmed cell death protein 1 inhibitors, clinical trials, biomarkers

## Abstract

**Simple Summary:**

Even with recent advances, there are urgent needs for novel therapies to improve overall survival and decrease toxicities in the management of head and neck squamous cell carcinoma (HNSCC). This article reviews historical data to provide a context and highlights recent data in understanding of epidemiology and pathophysiology and supporting changes in treatments of HNSCC, particularly in patients with recurrent and/or metastatic disease. For use of immune checkpoint modulators such as programmed cell death protein 1 (PD-1) inhibitors, potential predictive biomarkers of clinical benefits are also summarized. In addition, this article reviews currently ongoing clinical trials and provides a perspective on future research directions.

**Abstract:**

Head and neck squamous cell carcinoma (HNSCC) is the most common cancer arising in the head and neck region. The most common risk factors are smoking, excessive drinking, and human papillomavirus (HPV) infection. While the overall incidence of smoking is decreasing, the incidence of HPV-related HNSCC is increasing in the United States and Western Europe, which led to a shift in understanding of the pathophysiology, treatment, and prognosis of this disease. The outcomes for non-metastatic HNSCC remains very encouraging and continues to improve. Advances in radiation technology and techniques, better organ preserving surgical options, and multidisciplinary treatment modalities have improved cure rates for locally advanced HNSCC patients. The treatment of metastatic disease, however, remains an area of need. The advancement of immune checkpoint inhibitors has provided significantly better outcomes, but only a small proportion of patients obtain benefits. Most recurrent and/or metastatic HNSCC patients continue to have poor survival. This has led to the vigorous investigation of new biomarkers and biomarker-based therapies. Novel therapeutic options including adaptive cellular therapy and therapeutic vaccines are also on the horizon. In this review, we highlight the latest advances in the field of HNSCC and the future direction of research.

## 1. Introduction

Head and neck cancer is the seventh most common cancer worldwide, accounting for 3% of all cancers, with approximately 900,000 new cases and half a million deaths annually [1]. Among all cancers occurring in the head and neck region including oral cavity, oropharynx, hypopharynx, and larynx, the squamous cell carcinoma histology accounts for approximately 90% [2,3]. The major risk factors of head and neck squamous cell carcinoma (HNSCC) are tobacco and heavy alcohol use and human papillomavirus infection [4,5,6,7]. There has been a significant decline in smoking in high-income countries during the last few decades, which has led to a sharp decline in smoking related HNSCC [8,9]. In contrast, there has been a significant increase in global incidence of human papillomavirus (HPV)-associated or positive (+) HNSCC [4,10,11,12,13,14,15,16]. In addition to the shift in the common risk factors, we also have observed the shift in the management of HNSCC from indiscriminate intensification with a sole focus on improving survival to more personalized approaches based on understanding of the biology and leveraging advancements in biomarkers and immunotherapy. In this review, we will focus on the latest advances in the understanding and management of HNSCC and provide a perspective on future directions.

## 2. Epidemiology and Pathophysiology of HNSCC

### 2.1. HPV Positive (+) HNSCC

Human papillomavirus infection is now recognized as the major causative agent for HNSCC, especially in the oropharynx (OPSCC), accounting for approximately 60–70% of OPSCC in the United States, while the prevalence varies more within Western Europe ranging between 6.1 and 75% [16,17,18]. In third world countries, HPV(+) HNSCC is relatively rare with a <10% prevalence [19]. These variabilities are thought to be, at least in part, due to the different sensitivity and specificity of the HPV detection assays and differences in the study cohorts as well as different sexual practices and their associated risk factors in the study population [18]. HPV(+) HNSCC shows marked differences in epidemiology and pathophysiology as compared to HPV unrelated or negative (−) HNSCC [7,20,21]. Demographically, HPV(+) OPSCC patients tend to be younger males with a mean age of diagnosis in the 40–50’s and non-smokers or oligo-smokers [20,22]. HPV infection has a 10–30-year latency period between infection and clinical presentation with HPV(+) OPSCC [22]. Because HPV(+) OPSCC arises in the deep crypts in the tonsillar tissues without any associated pre-malignant clinical lesion within the oropharynx, early detection through screening is not possible [23]. In 2018, the FDA extended the approved age range of candidates for the preventive vaccine against HPV, GARDASIL 9, to include men and women <45 years of age. Epidemiological data suggest that prophylactic HPV vaccination reduces the prevalence of oral HPV infection by 88–93%. Considering the slow uptake and long latency period, vaccination is expected to reduce the incidence of oropharyngeal cancer by 2060, and we do not expect immediate changes in the current HPV(+) OPSCC incidence [24,25].

Regarding HPV-related pathophysiology, it is clearly established that HPV viral oncoproteins, E6 and E7, degrade two major tumor suppressors, p53 and pRb, upon infection, resulting in tumorigenesis in the reticular epithelium covering the tonsillar tissues, and persistent expression of E6 and E7 is required for tumor maintenance [26,27]. Disruption of the pRb function leads to a compensatory increase in expression of p16^INK4A^, which has been adapted as a surrogate marker of HPV infection in OPSCC [28]. The expression of p16 is now routinely tested using an immunohistochemistry staining as a standard of care in all OPSCC [29].

### 2.2. HPV Negative (−) HNSCC

The HPV(−) HNSCC is typically seen in patients with history of heavy tobacco and alcohol use [20,30]. There has been a significant decline in smoking in high-income countries during the last few decades, which has led to a sharp decline in smoking related HNSCC [8,9]. However, smoking-related cancers are still a significant problem in developing and third world countries [9]. In addition, HPV(−) HNSCC can occur in relatively young patients with no history of tobacco use, and the incidence has been rising with unclear etiology [31,32]. The most common genomic abnormalities in HPV(−) HNSCC from smokers are seen in *TP53* encoding p53 and *CDKN2A* encoding p16 with a distinct smoking signature, while the tumors from non-smokers have *TP53* mutations with aging and ultraviolet light exposure signatures [32,33,34].

## 3. Management of Newly Diagnosed, Locally Advanced HNSCC

The treatment of HNSCC requires a multidisciplinary approach. It has been established that such an approach in high volume centers produces consistently superior survival as well as improved short- and long-term toxicities [35]. Surgery, radiotherapy, and systemic therapies including chemotherapy, targeted therapy, and immunotherapy have a role in varying degrees at all stages of HNSCC. In general, patients with early stage disease defined as tumor stage, T1–2, and nodal stage, N0, at diagnosis are treated with surgery or radiotherapy as a single modality treatment and have an excellent outcome with cure rates of 70–90% [36]. However, locally advanced diseases with T3–4 and N1–3 at diagnosis usually require aggressive multi-modality therapies with a curative intent. Overall, the outcomes are excellent in non-smokers with HPV(+) HNSCC with a lower risk of death given standard of care therapies as compared to smokers with HPV(+) HNSCC or HPV(−) HNSCC [21,37,38].

### 3.1. Primary Surgical Approaches in HNSCC

It is a standard of care for HNSCC arising in the oral cavity to be treated with a primary surgical approach if a successful oncological resection is possible followed by adjuvant therapy in the form of radiation or concurrent chemoradiation depending on an assessment of high-risk features, as described in the next section [39]. Advanced stage laryngeal (e.g., T4a) and hypopharyngeal squamous cell cancers are also primarily treated with surgical intervention when functional preservation is not feasible using non-surgical approaches [39,40]. More recently, transoral robotic surgery (TORS) and transoral laser microsurgery (TLM) have emerged as the primary surgical modalities in the management of selected HNSCC. As compared to standard open surgery, these techniques have led to a 52% reduction in hospital stays after the surgery and a 90% reduction in feeding tube requirements at 1 year [41]. In OPSCC, TORS is primarily reserved for management of small to moderate size tumors (T1–T2) [42]. However, about 17–31% of patients who undergo surgery eventually need to receive radiation and/or chemotherapy in the adjuvant setting. Thus, about a quarter of these patients end up getting the triple modality therapy. This has led to an increasing shift, especially in high-volume centers, towards radiation alone or concurrent chemotherapy and radiation (CRT) for early stage OPSCC [43]. Similarly, the primary radiation-based approach is well established as a standard therapy for functional preservation in patients with HNSCC in the larynx and hypopharynx [44].

### 3.2. Post-Surgical Adjuvant Therapy

In the post-operative adjuvant setting, two landmark trials, EORTC 22931 and RTOG 9501, have established the standard of care delivering concurrent chemotherapy and radiotherapy (CRT) to HNSCC patients with a high-risk recurrence [45,46,47]. The EORTC 22931 trial showed that chemoradiotherapy improved progression-free survival (PFS), locoregional control (LRC), and overall survival (OS) compared to radiotherapy alone among patients deemed high-risk, which was defined as T3 or T4 disease, positive surgical margins, extra nodal spread, perineural or lymphovascular invasion, vascular tumor embolism, or those with oral cavity or oropharyngeal tumors with level IV or V nodes [45,47]. The RTOG 9501 trial, however, defined high-risk patients as patients with positive surgical margins, two or more involved regional nodes, or extra nodal extension [46,47]. Positive surgical margins and extracapsular extension were the only two risk features for which OS was positively impacted with CRT in both trials (EORTC 22931: *p* = 0.019; RTOG 9501: *p* = 0.063). There was also a trend in favor of CRT in the group of patients who had stage III–IV disease, perineural infiltration, vascular embolisms, and/or clinically enlarged level IV–V lymph nodes secondary to tumors arising in the oral cavity or oropharynx [47]. With the advent of immune checkpoint inhibitors (ICI), several studies are underway to evaluate the role of ICI in the adjuvant setting (ClinicalTrials.gov Identifiers: NCT02841748 and NCT02641093).

### 3.3. Non-Surgical Approach Using Concurrent Chemoradiation

Any surgery in the head and neck region requires special consideration for vital functions such as speech and swallowing as well as cosmesis. Therefore, locally advanced HNSCC are often treated using CRT for functional preservation. The meta-analysis of chemotherapy in head and neck cancer (MACH-NC study), resulted in the establishment of CRT as a standard of care for HNSCC when surgical resection is less feasible or would result in poor long-term functional outcomes. This analysis, which originally involved 17,346 patients with resectable [48] or unresectable, locally advanced HNSCC, was updated to involve 19,248 patients [49]. These meta-analyses together confirmed that the addition of concurrent chemotherapy with radiotherapy increased OS with an absolute benefit of 6.5% at five years (hazard ratio for death: 0.83; 95% CI: 0.79 to 0.87; *p* < 0.001) and decreased locoregional failure rates with CRT as compared to local therapy alone.

It also established the beneficial effect of the timing of adding chemotherapy that concurrent administration of chemotherapy and radiation improved OS, while the adding chemotherapy before or after radiation as induction or adjuvant therapy did not improve OS [48,49]. Similarly, the RTOG 91-11 trial established concurrent CRT as the most effective approach for locoregional control and organ preservation, in patients with resectable stage III or IV glottic or supraglottic disease, compared to radiation alone or induction chemotherapy followed by radiation [50]. A Phase 3 trial comparing induction chemotherapy with docetaxel, cisplatin, and 5-fluorouracil (TPF) followed by radiation in responders versus TPF followed by concurrent CRT in T3 and T4 SCC of larynx and hypopharynx requiring total laryngectomy (SALTORL trial. NCT03340896) is currently ongoing. The primary endpoint of the SALTORL trial is laryngo-esophageal dysfunction-free survival as a composite endpoint.

For the management of persistent or recurrent disease in either the primary site or the regional lymph nodes after CRT, surgery is preferred as a salvage therapy. The five-year disease-specific survival after salvage surgery has been shown to be approximately 55% [51]. Therefore, salvage surgeries are reserved for patients with recurrent or persistent disease or severe functional impairment after the function preservation approach with radiation alone or CRT. For patients with unresectable, locally advanced disease, sequential therapies using induction chemotherapy followed by CRT have been explored as a treatment intensification approach [52,53,54]. This patient population has a particularly poor prognosis requiring improved treatments for better OS. However, when the induction chemotherapy with TPF followed by CRT with a platinum agent was compared to CRT alone, there was no clear improvement in OS [52,53,54]. The role of induction chemotherapy has continued to be evaluated as a way to select patients who can safely receive de-intensified radiation or CRT based on their response to upfront induction chemotherapy [55,56].

### 3.4. Choice of Systemic Therapy during Concurrent Chemoradiation

For patients with locally advance HNSCC, a high-dose cisplatin (100 mg/m^2^, administered intravenously every 21 days for three cycles) given concurrently with radiotherapy is the standard of care with established OS benefits [48,57]. However, due to the significant toxicities, a low-dose weekly cisplatin (30–40 mg/m^2^, administered intravenously every 7 days for the duration of radiation therapy) is frequently used in both definitive and adjuvant CRT settings, because the weekly regimen offers several advantages over the high-dose cisplatin including ease of administration and reduced toxicity [58]. A retrospective study as well as prospective Phase II/III studies comparing the efficacy of the 40 mg/m^2^ cisplatin weekly and 100 mg/m^2^ cisplatin every 3 weeks demonstrated non-inferiority of the weekly regimen for survival and locoregional control rates with fewer incidences of neutropenia, nephrotoxicity, and ototoxicity with low-dose cisplatin [59,60,61]. However, we need to wait for sufficient maturity of these data before suggesting changes to the current standard of care.

In patients who are unable to receive platinum-based chemotherapy, cetuximab, an epidermal growth factor receptor chimeric IgG1 monoclonal antibody, was shown to improve survival in patients with locoregional advanced HNSCC as compared to radiation alone leading to its approval for use concurrently with radiation [62]. Cetuximab was initially thought to be less toxic given with radiation compared to cisplatin and tested as a less toxic alternative to cisplatin in deintensification trials for HPV(+) HNSCC with a favorable prognosis as further discussed in the next section. However, recently, two randomized trials, the De-Escalate trial and the Phase III RTOG 1016 trial comparing radiotherapy plus cetuximab versus radiotherapy plus cisplatin in patients with HPV(+) OPSCC, have shown worse outcomes when given cetuximab compared to cisplatin [63,64].

With a recent advancement in immune checkpoint inhibitors (ICI), there are many trials underway to evaluate the safety and efficacy of ICI combined with radiation in HNSCC. There are two programmed death 1 (PD-1) inhibitors, pembrolizumab and nivolumab, which are approved for use in HNSCC [65,66]. Currently, we have robust data regarding safety, but efficacy data are scant in management of locally advanced HNSCC. Pre-clinical studies clearly demonstrate that radiation modulates the immune system in ways that can augment treatment responses when combined with immunotherapy. Studies have demonstrated excellent tolerance and lower incidence of Grade 3 rash and mucositis in platinum ineligible patients who received pembrolizumab with radiation [67]. Studies in locally advanced HNSCC using pembrolizumab or nivolumab with cisplatin and radiation were shown to be safe without unexpected toxicities [68,69]. However, the results of the Phase III JAVELIN Head and Neck 100 trial evaluating avelumab (PD-L1 inhibitor) plus CRT followed by avelumab maintenance versus CRT in patients with locally advanced HNSCC did not show any survival benefits (PFS stratified HR 1.21, 95% CI 0.93–1.57; stratified *p*-value 0.92 and OS stratified HR 1.31, 95% CI 0.93–1.85; stratified *p*-value 0.94; ClinicalTrials.gov Identifier: NCT02952586) [70]. The use of ICI as well as other targeted agents as maintenance therapy after CRT (ClinicalTrials.gov Identifier: NCT03811015 and NCT00079053) is under investigation.

### 3.5. Deintensification Efforts in Management of Low-Risk HPV(+) OPSCC

The Danish Head and Neck Cancer Group 5 study demonstrated that p16 positivity used as a surrogate marker of HPV positivity was associated with a 5-year LRC rate of 58% and OS of 62% [71]. In the RTOG 0129 study, HPV(+) OPSCC was associated with a significantly more favorable prognosis than HPV(−) OPSCC given CRT (3-year OS 82.4% versus 57.1%; *p* < 0.001) [21]. Although patients with HPV(+) OPSCC typically presented with small primary tumors, they tend to have cervical lymph node involvement [72] and tend to have improved outcomes compared to HPV(−) HNSCC (5-year OS of 80–87% for patients with N1–N2c HPV(+) OPSCC versus 37–58% for those with N1–N2c HPV(−) OPSCC) [73]. The staging groups defined by the American Joint Committee on Cancer (AJCC) in 2010 [74] did not reflect the real prognostic differences between HPV(+) and HPV(−) HNSCC [72]. As a result, the eighth edition of the AJCC staging manual separated the staging system of HNSCC into HPV(+) and HPV(−) subgroups to accurately reflect the differences in their prognosis [75]. As a result of the new staging system, 92% of patients with HPV(+) OPSCC could be down staged, and up to 64% of patients were now staged as stage I disease, up from 3% in the old staging system [76].

Since HPV(+) OPSCC has a favorable prognosis and current treatment regimens are associated with significant morbidities, there is ongoing interest to investigate whether current treatments could be de-intensified with similar levels of disease control but with fewer acute and/or chronic toxicities. Deintensification is considered the most appropriate in patients with HPV(+) OPSCC with the lowest risk of disease recurrence such as those with a smoking history of <10 pack year and T1–T3 primary tumors [77]. The foremost current strategy is to omit or substitute concurrent cisplatin given with radiotherapy with less toxic agents such as PD-1 inhibitors (NCT02764593) to reduce the dose and/or field of radiation (NCT02254278) and to use induction chemotherapy response as a selection criteria to follow the treatment by a drastic reduction in radiation dose (NCT01706939). Studies are ongoing and current data are insufficient to recommend any de-intensified treatment for HPV(+) HNSCC.

### 3.6. Post-Chemoradiation Response Assessment

The role of elective neck dissection after definitive CRT has been evaluated in a Phase III non-inferiority trial where 564 HNSCC patients with advanced nodal stages were randomly assigned to receive definitive CRT followed by either elective neck dissection (END) within 4–8 weeks or PET-CT scans at 12 weeks [78]. The 2-year OS rate was 84.9% in the surveillance group and 81.5% in the planned-surgery group. PET-CT-guided surveillance resulted in fewer neck dissections than planned neck dissection surgery (54 vs. 221); therefore, it had fewer surgical complications (7.8% versus 29.4%). The PET-CT-guided surveillance was also cost effective. Thus, PET-CT-guided surveillance after CRT remains standard of care.

## 4. Management of Recurrent and/or Metastatic HNSCC

Overall, more than 65% of patients develop recurrent and/or metastatic (R/M) HNSCC, and the majority are considered incurable given palliative chemotherapies [79]. However, recent advancement in immunotherapy resulted in a paradigm shift in the management of incurable HNSCC (Table 1). Biomarker study and evaluating factors influencing outcomes of treatment is also being vigorously investigated.

### 4.1. Advances in Immune Checkpoint Inhibitors

Before the advent of ICI, the Phase 3 Erbitux in First-Line Treatment of Recurrent or Metastatic Head and Neck Cancer (EXTREME) trial established cetuximab added to chemotherapy consisting of 5-fluorouracil plus a platinum (cisplatin or carboplatin) as the first-line standard of care therapy. The EXTREME trial showed that the three-drug combination had a superior OS rate (36 vs. 20%), significantly improved median OS (10.1 vs. 7.4 months; hazard ratio for death, 0.80; 95% CI, 0.64 to 0.99; *p* = 0.04) as well as improved median PFS (5.6 vs. 3.3 months) compared to chemotherapy alone [80]. However, the development of the PD-1 inhibitors and other immune-checkpoint inhibitors has greatly changed the treatment of HNSCC. The PD-1 inhibitors, pembrolizumab and nivolumab, were the earliest drugs to show durable responses and improved survival in patients with R/M HNSCC, leading to approval by the Food and Drug Administration (FDA). The Phase Ib trial, KEYNOTE-012, in tumors with PD-L1 expression >1% and the expansion cohort of the same study showed durable responses to pembrolizumab (18% response rate with median OS of 8 months), leading to its accelerated FDA approval [81,82]. Soon after, nivolumab was approved in the R/M HNSCC population based on the data from the Phase III CheckMate 141 trial, which showed a 13% response rate and improved OS (2 year OS of 17 vs. 6%) and significantly lower grade 3–4 adverse events compared to the investigators’ choice of standard-of-care systemic therapy [66,83].

This was followed by a Phase 3 KEYNOTE-048 study, which randomly assigned 882 untreated patients with R/M HNSCC to treatment in the first line setting with pembrolizumab alone, pembrolizumab with chemotherapy (5-fluorouracil and platinum), or the standard regimen of 5-fluorouracil and platinum plus cetuximab (the regimen in the EXTREME trial) [84]. Both pembrolizumab monotherapy and pembrolizumab combined with chemotherapy improved the primary end point of OS in patients with PD-L1–expressing tumors (both combined positive score cutoffs of 20 or higher and 1 or higher) when separately compared with the three-drug EXTREME regimen. The response rates were lower (17%) with pembrolizumab alone but was at par with EXTREME regimen when pembrolizumab was combined with chemotherapy (36%). There were fewer grade 3–4 adverse effects with pembrolizumab alone when compared with the EXTREME regimen (17 vs. 69%) [84]. On the basis of KEYNOTE-048, pembrolizumab monotherapy or in combination with chemotherapy has supplanted the EXTREME regimen as the first line therapy in patients with R/M HNSCC, especially in PD-L1 expressing tumors. Recently, promising data with median OS up to 21.9 months have been demonstrated in patients receiving docetaxel, cisplatin, and cetuximab (TPExtreme) followed by a PD-1/PD-L1 inhibitor in the second line setting [85]. Though we do not have data on PD-L1 expression status and still await the final analysis, this regimen could be a promising option for patients without PD-L1 expression.

### 4.2. Current Biomarkers of Response for Immune Checkpoint Inhibitors

Successful development of targeted therapies in biomarker-selected patients for personalized medicine has shifted the expectations in cancer research, but the lack of targetable genomic abnormalities in HNSCC limited the development of targeted therapies in the past [33]. However, immune checkpoint inhibitors (ICI) have ushered in a new era and revolutionized the therapy of all cancers including HNSCC. Despite the initial enthusiasm, the clinical benefits of PD-1/PD-L1 ICI in R/M HNSCC patients are overall limited [66,81,84]. In this context, it is important to look at biomarkers that can predict the response and durability of clinical benefits provided by these therapies, and better tailor treatments for individual patients. Evidence is accumulating to show that several tumor and host factors shape the response to ICI therapy.

#### 4.2.1. Programmed Death-Ligand 1 (PD-L1)

PD-L1 expression was the first to be studied as a predictive biomarker of PD-1/PD-L1 inhibitors. PD-L1 expression on immune cells in pre-treatment tumor biopsies has been associated with improved treatment outcomes [86]. Further studies in HNSCC tumors showed that the number of PD-L1 positive cells including tumor, lymphocytes, and macrophages, in relation to total tumor cells, also known as combined positive score (CPS), is a more predictive biomarker of response than the measurement of PD-L1 expression on tumor cells alone [87]. Testing PD-L1 expression and assessment of CPS are now a standard of care as a part of the R/M HNSCC management and decision to pembrolizumab monotherapy versus pembrolizumab and chemotherapy.

#### 4.2.2. Tumor Mutational Burden (TMB)

TMB is being evaluated as a biomarker of ICI response in multiple cancers. Increased TMB has been related to improved response to immunotherapy in non-small cell lung cancer (NSCLC) [88] and clinical response and OS in melanoma [89]. The association of TMB in HNSCC has produced mixed outcomes. While KEYNOTE-012 demonstrated a positive correlation with pembrolizumab response and total mutational load using a cutoff of ≥102 mutations per exome [81], two other studies concluded that TMB had no correlation with immune cell infiltrates evaluated by analyzing the RNA sequencing data from The Cancer Genome Atlas (TCGA) HNSC and the Chicago Head Neck Genomics [90,91]. TMB testing in HNSCC is not yet recommended as a standard of care [87].

#### 4.2.3. Tumor Immune Microenvironment

Assessment of the tumor immune microenvironment (TIM) is under vigorous research to identify potential biomarkers for ICI response. Increased infiltration of CD3^+^CD8^+^ and CD3^+^Foxp3^+^ tumor-infiltrating lymphocytes (TILs) are associated with better prognosis in HNSCC [92]. In addition, higher CD8^+^ T-cell infiltration has been observed among PD1/PD-L1 inhibitor therapy responders and has been proven to be an independent predictive factor for improved prognosis [93]. At the same time, the presence of myeloid-derived suppressor cells (MDSCs), IL-1 and IL-6 expression from cancer cells, M2-polarized tumor-associated macrophages (TAMs), and N2 tumor-associated neutrophils (TANs) are associated with attenuated response to ICI therapy [91]. Pooled analysis of tissue samples from KEYNOTE-012 and KEYNOTE-055 has shown that PD-L1 expression, 18-gene T-cell-inflamed signature, and TMB had a positive correlation with response and survival when treated with single agent pembrolizumab [94,95].

#### 4.2.4. Human Papillomavirus Status

HPV(+) HNSCC has a favorable prognosis and survival even in R/M HNSCC patients given a standard of care [38,96,97]. However, the value of HPV as a predictor of ICI response remains to be determined. Although PD-L1 expression appears to be unrelated to HPV status, HPV(+) HNSCC has increased intra-tumoral CD8^+^ T-cells and Treg/CD8^+^ ratio [87]. Conversely, HPV(−) HNSCC tumors do not have a high immune infiltration but tend to have a higher TMB. Thus, the overall impact of HPV status on ICI response is unpredictable and clinical trials so far have failed to show a clear association of HPV status with response to PD-1 inhibition therapy [81,98,99].

#### 4.2.5. Oral and Gut Microbiome

Microbiome is emerging as a major factor influencing response to cancer therapy and has the potential for modulation to improve responses given immunotherapy. The role of the oral cavity microbiome is being vigorously investigated in HNSCC development and progression, and initial studies have pointed to a possible role of Fusobacteria species, which have been found in abundance in both primary and metastatic cancerous tissues [100]. Oral microbiome has the potential to affect a toxicity profile in patients undergoing concurrent CRT. Preliminary studies have shown that oral bacterial alpha diversity has been shown to correlate with the severity of mucositis and candidiasis [101]. The role of host gut microbiota and its modulation of immune system through changes in metabolomics and subsequent effect on ICI therapy outcomes are being studied. Several different genera of gut microbes including *Akkermansia*, *Fecalibacterium*, *Bifidobacterium*, etc., have shown association with response to ICI therapy. Microbiome diversity was also greater among responders, and fecal microbiota transplant (FMT) of these organisms into germ-free mice led to increased responses [102]. Studies in HNSCC are currently ongoing.

## 5. Future Direction in Clinical Research

Unfortunately, an estimated 70–90% of patients with R/M HNSCC have no response to ICI, or initial responses are followed by disease progression, ultimately leading to death from the disease. To improve response rates and survival, ongoing trials are evaluating combinations involving ICIs, therapeutic vaccines, co-stimulatory agonists, and cytotoxic agents (Table 2 and Figure 1).

### 5.1. Immunotherapy Combinations for Metastatic HNSCC

The combination of PD-1 and CTLA-4 inhibitors (e.g., nivolumab and ipilimumab) has shown a synergistic effect and efficacy in melanoma [104] and other cancers. The Phase II CONDOR study evaluated a combination of CTLA-4 and PD-L1 inhibitors, tremelimumab and durvalumab, in R/M HNSCC patients with no or low (<25%) PD-L1 expression in the tumor cells. Unfortunately, the trial showed discouraging results with an overall response rates of 8 and 9% being observed with the combination and durvalumab monotherapy, respectively [105]. Additional Phase III studies are ongoing to determine the efficacy of combination therapy approaches (NCT02551159, NCT02369874). In addition, there are several HPV-directed therapies combining with PD-1 inhibitors. One example is using a novel fusion protein (CUE-101) designed to activate and expand a population of tumor-specific T cells to eradicate HPV-driven cancer cells with or without a PD-1 inhibitor (NCT03978689).

### 5.2. Therapeutic Vaccines

Another area of active research involves the use of therapeutic vaccines, and multiple Phase I/II trials in HNSCC patients are underway. Some of the completed trials have shown encouraging results. The adjuvant peptide-loaded dendritic cell vaccine against p53 was tested in a Phase I study. Two-year disease-free survival was 88% and decreased Treg levels, and modest vaccine-specific immunity was also seen [106]. The Phase I/II study to evaluate talimogene laherparepvec in combination with cisplatin and radiotherapy, in the setting of locoregionally advanced HNSCC, demonstrated 100% locoregional disease control, a 76% relapse free rate and an OS of 70.5% at 29 months [107]. A Phase II study of a combination of nivolumab and a synthetic long-peptide HPV-16 vaccine (ISA101—a synthetic long-peptide HPV-16 vaccine inducing HPV-specific T cells) showed promising results with an overall response rate of 33% and a median OS of 17.5 months, both of which are higher than historically observed with immunotherapy alone [108]. A Phase 2 clinical trial using cemiplimab with ISA101b is being planned (NCT04398524).

### 5.3. Adaptive Cellular Therapy

Cell therapy-based options are the latest weapon in the arsenal to fight HNSCC. This involves tumor cell death induced by activated cytotoxic T lymphocytes (CTLs) and involves use of T cells already primed to patient-specific antigens. Chimeric antigen receptor T cells (CAR-T) is one such therapy that has been successful in hematologic malignancies, and multiple ongoing clinical trials are investigating the use of T-cell therapies in solid malignancies as well [109]. A previous study to evaluate efficacy of adoptive immunotherapy using ex-vivo-activated CTLs in the treatment of five patients with advanced HNSCC has shown some promise with 100% 1-year OS in responders [110]. In HNSCC, the ongoing trials include use of E7 T-cell receptor (TCR) T cells (KITE-439) in HPV 16+ cancers (NCT03912831), use of adoptive T cell transfer in HPV-mediated disease (NCT03578406), and autologous tumor infiltrating lymphocytes (TIL) infusion (LN-145/LN-145-S1) followed by IL-2 in patients with R/M HNSCC (NCT03083873).

## 6. Conclusions

The management of HNSCC is rapidly evolving. Continued technological advances in surgery and radiotherapy as well as use of concurrent systemic therapies have contributed to significant improvements in outcomes for patients with non-metastatic disease. However, this still comes with significant toxicities. Furthermore, the outcome for R/M HNSCC remains poor for most patients. Although immunotherapy has been able to show durable responses, this benefit is seen in only a limited number of patients. Investigational strategies using immunotherapy, vaccines, cellular therapy, and optimization of incorporation of biomarkers promise to further advance the field.

## Figures and Tables

**Figure 1 cancers-13-00338-f001:**
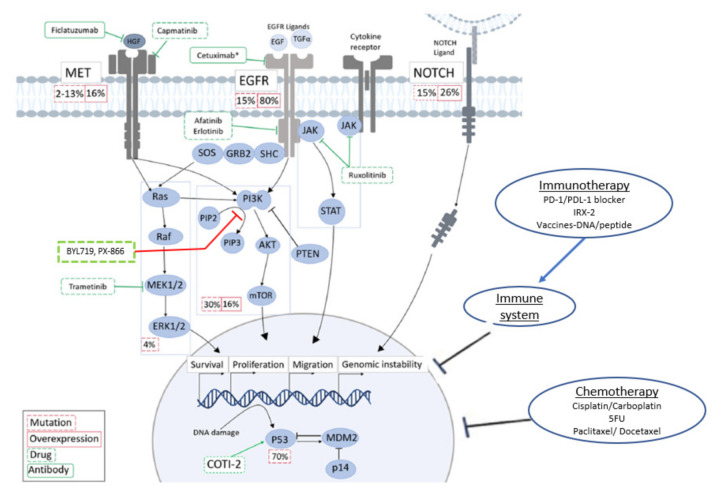
Major pathways dysregulated in HNSCC with approved and investigational therapeutic agents and their sites of activity. Derivative of original figure from Alsahafi et al. [103] licensed under CC BY 4.0 https://creativecommons.org/licenses/by/4.0/. HGF: hepatocyte growth factor, EGFR: epidermal growth factor receptor, JAK: Janus kinase, PIP: prolactin induced protein, ERK: extracellular-signal-regulated kinase, PTEN: Phosphatase and tensin homolog, STAT: signal transducer and activator of transcription, MDM2: Mouse double minute 2 homolog, mTOR: mechanistic target of rapamycin

**Table 1 cancers-13-00338-t001:** Results of immune checkpoint inhibitor trials in head and neck squamous cell carcinoma (HNSCC).

Relapsed/Metastatic Pretreated HNSCC				
Study	Phase	Treatment	Response	Overall Survival
CheckMate 141	3	Nivolumab vs. investigator’s choice	13 vs. 6%	17 vs. 6% at 2 years
KEYNOTE 012	1b	Pembrolizumab	18%	Median: 8 months
KEYNOTE 055	2	Pembrolizumab	16%	Median: 8 months
KEYNOTE 040	3	Pembrolizumab vs. investigator’s choice	15 vs. 10%	Median: 8.4 months vs. 6.9 months
First-line therapy in metastatic HNSCC				
KEYNOTE 048	3	Pembrolizumab vs. pembrolizumab + chemo vs. EXTREME regimen	17 vs. 36 vs. 36%	Pembrolizumab vs. EXTREME regimen*: 14.9 vs. 10.7 months for CPS > 20, 12.3 vs. 10.3 months for CPS > 1 Pembrolizumab + chemo vs. EXTREME regimen: 13 vs. 10.7 months
PDL-1 plus CTLA-4 blocker combination in pretreated metastatic HNSCC				
CONDOR	2	Durvalumab + Tremelimumab vs. Durvalumab monotherapy vs. Tremelimumab monotherapy	8 vs. 9 vs. 2%	7.6 vs. 6 vs. 5.5 months
CheckMate 651	3	Nivolumab + Ipilimumab vs. EXTREME regimen	Results awaited	Results awaited
CheckMate 714	2	Nivolumab + Ipilimumab vs. Nivolumab	Results awaited	Results awaited

* EXTREME regimen: Phase 3 Erbitux in First-Line Treatment of Recurrent or Metastatic Head and Neck Cancer Trial—Regimen containing 5-fluorouracil + platinum (cisplatin or carboplatin) +/− cetuximab.

**Table 2 cancers-13-00338-t002:** Ongoing clinical trials in HNSCC.

Immune Checkpoint Blocker Trials in Metastatic HNSCC			
Study Name	Phase	NCT	Treatment Regimen
KEYNOTE B10	4	NCT04489888	Pembrolizumab + Carboplatin + Paclitaxel
LEAP-10	3	NCT04199104	Pembrolizumab vs. Pembrolizumab + Lenvatinib
	1	NCT03498378	Avelumab + Palbociclib + Cetuximab
	1 and 2	NCT03650764	Pembrolizumab + Ramucirumab
	1 and 2	NCT03655444	Nivolumab + Abemaciclib
	2	NCT04220866	Intratumoral MK-1454 + Pembrolizumab vs. Pembrolizumab
	1b/2	NCT04193293	Duvelisib + Pembrolizumab
Vaccine and Cellular Therapy trials in metastatic HNSCC			
	2	NCT04369937	ISA101b + Pembrolizumab + Cisplatin chemo RT
	1	NCT04290546	CIML NK cells + IL-14 ± Ipilimumab
	2	NCT03083873	Autologous TIL (LN-145/LN-145-S1)
	1	NCT03247309	TCR-engineered T Cells in Solid Tumors (ACTengine IMA201-101) (ACTengine)
	1	NCT03912831	HPV16 E7 T Cell Receptor Engineered T Cells (KITE-439) in HLA-A*02:01 + Subjects

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
