# Peer review of "Recent Advances and Future Directions in Clinical Management of Head and Neck Squamous Cell Carcinoma"

_cancers, 2021, doi:10.3390/cancers13020338_

Round 1

Reviewer 1 Report

This article was an interesting read. The manuscript is generally well written and clearly presented.

It also addresses a subject that is of great interest to the scientific community.

I would only ask the authors to rephrase the abstract as it coincides with the introduction.

Author Response

Reviewer 1:

Only ask the authors to rephrase the abstract as it coincides with the introduction.

Answers: Abstract has been revised.

Reviewer 2 Report

Page 2 : HPV + OPSCC : in western Europe incidence seems to bo less than incidence in the US, 60-70%, and there is heterogeneity from one country to an other. Could the authors comment on that ?

Page 3 : for medical approach in organ préservation the authors mentionned concurrent chemoradiotherapy, but not the sequential approach with induction chemotherapy followed by XRT. Could the authors comment on that ?

Page 4 : the RTOG trial demonstrated that concurrent chemoradiotherapy is better than sequential treatment when PF is used as induction chemo. Pointreau Y et al demonstrated in GORTEC trial that TPF is Superior to PF for laryngeal preservation, when induction chemotherapy is used. There is an ongoing trial comparing concurrent hemoradiotherapy to TPF induction + XRT (SALTORL trial)

Page 4 : for systemic chemotherapy during XRT, high dose cisplatinum remains the standard of care, weekly Cisplatinum is not as effective as 3 weekly high dose especially in adjuvant setting, recently presented data (in ESMO) showed that weekly cisplatinum seems to be not inferior but data are not sufficiently mature to conclude.

Page 7 : on the KEYNOTE 048 trial, do the authors consider that Pembrolizumab monotherapy and pembrolizumab + chemo should be given to patients with tumors with CPS > 1 ? How do we have to select patients for pembro mono or Pembro + chemo combi ? What about patients with CPS < 1, is there still a place for chemo without ICI ? Could the authors comment on the results of the TPEx trial, présented at ASCO 2020 by Guigay J et al ?

We would like to congratulate authors for this excellent work.

Author Response

Reviewer 2:

Page 2 : HPV + OPSCC : in western Europe incidence seems to bo less than incidence in the US, 60-70%, and there is heterogeneity from one country to another. Could the authors comment on that?

Answers: Heterogeneity in the incidence of HPV-related OPSCC in different countries has been commented in Section 2.1 as “…while the prevalence varies more within Western Europe ranging 6.1%-75% [16-18]. In third world countries, HPV(+) HNSCC is relatively rare with a <10% prevalence [19]. These variabilities are thought to be, at least in part, due to the different sensitivity and specificity of the HPV detection assays and differences in the study cohorts as well as different sexual practices and their associated risk factors in the study population [18].” Reference has been updated.

Page 3 : for medical approach in organ preservation the authors mentioned concurrent chemoradiotherapy, but not the sequential approach with induction chemotherapy followed by XRT. Could the authors comment on that ?

Answers: The comparison of concurrent and sequential approach to chemotherapy and radiation has been added as, “For patients with unresectable, locally advanced disease, sequential therapies using induction chemotherapy followed by CRT have been explored as a treatment intensification approach [52-54]. This patient population has a particularly poor prognosis requiring improved treatments for better OS. However, when the induction chemotherapy with TPF followed by CRT with a platinum agent was compared to CRT alone, there was no clear improvement in OS [52-54]. The role of induction chemotherapy is continued to be evaluated as a way to select patients who can safely receive de-intensified radiation or CRT based on their response to upfront induction chemotherapy [55,56].”

Page 4: the RTOG trial demonstrated that concurrent chemoradiotherapy is better than sequential treatment when PF is used as induction chemo. Pointreau Y et al demonstrated in GORTEC trial that TPF is Superior to PF for laryngeal preservation, when induction chemotherapy is used. There is an ongoing trial comparing concurrent chemoradiotherapy to TPF induction + XRT (SALTORL trial).

Answers: We have added the following, “ A phase 3 trial comparing induction chemotherapy with docetaxel, cisplatin, and 5-fluorouracil (TPF) followed by radiation in responders versus TPF followed by concurrent CRT in T3 and T4 SCC of larynx and hypopharynx requiring total laryngectomy (SALTORL trial. NCT03340896) is currently ongoing. The primary endpoint of the SALTORL trial is laryngo-esophageal dysfunction-free survival as a composite endpoint.”

Page 4 for systemic chemotherapy during XRT, high dose cisplatinum remains the standard of care, weekly Cisplatinum is not as effective as 3 weekly high dose especially in adjuvant setting, recently presented data (in ESMO) showed that weekly cisplatinum seems to be not inferior but data are not sufficiently mature to conclude.

Answers: Thought the data for non-inferiority of weekly cisplatin is promising, we agree that data are not sufficiently mature. We have added the following to reflect that “However, we need to wait for sufficient maturity of these data before suggesting changes to the current standard of care.”

Page 7 : on the KEYNOTE 048 trial, do the authors consider that Pembrolizumab monotherapy and pembrolizumab + chemo should be given to patients with tumors with CPS > 1 ? How do we have to select patients for pembro mono or Pembro + chemo combi ? What about patients with CPS < 1, is there still a place for chemo without ICI ? Could the authors comment on the results of the TPEx trial, présented at ASCO 2020 by Guigay J et al ?

Answers: We have added the following line “Recently, promising data with median OS up to 21.9 months has been demonstrated in patients receiving docetaxel, cisplatin, and cetuximab (TPExtreme) followed by a PD-1/PD-L1 inhibitor in the second line setting [85]. Though we do not have data on PD-L1 expression status and still await the final analysis, this regimen could be a promising option for patients without PD-L1 expression.” Reference has been added.

Reviewer 3 Report

Recent Advances and Future Directions in Clinical Management of Head and Neck Squamous Cell Carcinoma

The authors presented a review, which focus on the latest advances in the understanding and management of head and neck squamous cell carcinoma (HNSCC).

This paper has several strong points that I can mention:

  • The subject of this research is interesting and it will make a lots of readers interested and absorbed to this manuscript;
  • The manuscript is made in a good structure and authors managed to mention their points clearly.

However, there are several points that must be corrected before the manuscript goes for publication:

  • A major point about this paper would concern lack of discussions on using computer aided diagnosis and prognosis. There are many great works out there and this review limited itself on about 100 references which can be far more.
  • Another point can be related to more explicit description for each subsection. Paragraphs are rather too short and less connected.
  • There is need to provide much better representation of this diseases with underlying causes.

A minor editorial comment:

Please make your graph more adjusted with high quality. They are readable but can be made with higher quality.

In overall, the manuscript discusses a very intriguing survey. Therefore, I recommend major revision.

Thank you

Author Response

Reviewer 3:

Concern lack of discussions on using computer aided diagnosis and prognosis. There are many great works out there and this review limited itself on about 100 references which can be far more.

Answers: We appreciate your comment. However, we feel it is not within the scope of this review.

More explicit description for each subsection. Paragraphs are rather too short and less connected. Provide much better representation of this diseases with underlying causes.

Answers: We appreciate your comment. However, for busy clinicians and clinical researchers who do not have a lot of time to read a very long review, we feel succinct descriptions of the current landscape are also valuable.

Please make your graph more adjusted with high quality. They are readable but can be made with higher quality.

Answers: We reproduced the figure from previously published work. Unfortunately, we do not have a higher resolution figure. We hope the current journal, Cancers, may be able to enhance the figure during their editorial process.

Round 2

Reviewer 3 Report

I copied and pasted my comments and authors replies. The replies were in overall like: thank you for your comment BUT....

  • I asked for a discussion on potential application of CADs in medicine, no scientific response
  • I asked the paragraphs are too long and not connected, authors bragged about being a MDs too busy to do science!
  • I even asked the simplest thing to make your figure better, they left this important duty of authors to the journals

It is really unfortunate that I received such responses as I thought that this paper might have some potential, but I can not accept this paper due to lack of any scientific arguments.

Thank you

Concern lack of discussions on using computer aided diagnosis and prognosis. There are many great works out there and this review limited itself on about 100 references which can be far more.

Answers: We appreciate your comment. However, we feel it is not within the scope of this review.

More explicit description for each subsection. Paragraphs are rather too short and less connected. Provide much better representation of this diseases with underlying causes.

Answers: We appreciate your comment. However, for busy clinicians and clinical researchers who do not have a lot of time to read a very long review, we feel succinct descriptions of the current landscape are also valuable.

Please make your graph more adjusted with high quality. They are readable but can be made with higher quality.

Answers: We reproduced the figure from previously published work. Unfortunately, we do not have a higher resolution figure. We hope the current journal, Cancers, may be able to enhance the figure during their editorial process.